# HLA-G and Recurrent Pregnancy Loss

**DOI:** 10.3390/ijms24032557

**Published:** 2023-01-29

**Authors:** Greta Barbaro, Annalisa Inversetti, Martina Cristodoro, Carlo Ticconi, Giovanni Scambia, Nicoletta Di Simone

**Affiliations:** 1Dipartimento di Scienze della Salute della Donna, del Bambino e di Sanità Pubblica, Fondazione Policlinico Universitario A. Gemelli, Istituto di Ricovero e Cura a Carattere Scientifico (I.R.C.C.S.), L. go A. Gemelli 8, 00168 Roma, Italy; 2Department of Biomedical Sciences, Humanitas University, Via Rita Levi Montalcini 4, Pieve Emanuele, 20072 Milano, Italy; 3IRCCS Humanitas Research Hospital, Via Manzoni 56, 20089 Rozzano, Italy; 4Dipartimento di Scienze Chirurgiche, Ginecologia e Ostetricia, Università di Torvergata, 00168 Roma, Italy; 5U.O.C. di Ginecologia Oncologica, Fondazione Policlinico Universitario A. Gemelli IRCCS, L. go A. Gemelli 8, 00168 Rome, Italy

**Keywords:** HLA-G, recurrent pregnancy loss, trophoblast, placentation

## Abstract

Placentation is an immunological compromise where maternal immune system cells and trophoblastic cells interact to reach an equilibrium condition. Although the cross talk between the two systems is complex and not completely understood, Human Leukocyte Antigen G (HLA-G), expressed on trophoblastic cell surfaces, seems to be one of the main molecules involved in the modulation of both local and systemic maternal immune response. The prevalence of recurrent pregnancy loss (RPL), probably underestimated, is 5% of all women who achieve pregnancy, and about 40–60% percent of RPL cases are unexplained. There is an immunological analogy between allograft rejection and miscarriage, and the purpose of this review is to describe how the HLA-G pathway alterations are involved in disrupting the immunologic balance and in increasing the risk of recurrent pregnancy loss.

## 1. Introduction

Human Leukocyte Antigen G (HLA-G) is a molecule that has been known for half a century. For the first time, in 1990, it was discovered that HLA-G was expressed on trophoblast cell surfaces and was identified as a molecule inducing an immunotolerance state in pregnancy [1]. At first, the HLA-G role was confined to the feto–maternal interface, but over the years, the interest in this molecule has begun to increase in several medical fields. In fact, the physio-pathological mechanisms involving HLA-G have been widely investigated and described in transplant medicine, rheumatology, oncology and infectious disease. As a tolerogenic molecule, HLA-G seems to be protective in presence of inflammation or in autoimmune diseases, whereas conversely, HLA-G allows tumoral disease progression, inducing a state of immunotolerance for cancer cells where it can be neo-expressed and promote cancer progression [2]. For example, in the context of Merkel-cell carcinoma induced by Merkel cell polyomavirus, the loss of HLA-I is described as a pathogenetic mechanism leading to escape from the immunological response against the tumor [3]. Over the years, knowledge about the HLA-G molecule’s role in reproductive medicine and obstetrics has grown. Precision medicine is becoming the target in different fields, and an HLA-G gene mutation test fits the need of personalized medicine; however, the interpretation of the results can be difficult and might require a basic understanding of genetics and immunology, often far from daily clinical practice. Recurrent pregnancy loss (RPL), defined as the loss of two or more pregnancies [4], is a distressing pregnancy disorder affecting 2–5% of couples trying to conceive [5]. The pathophysiological mechanisms of RPL are largely unknown, although several risk factors have been associated with RPL. Among these, maternal age plays a key role: women older than 40 years present a higher risk of pregnancy loss [6]. Furthermore, genetic alterations, such as chromosomal abnormalities, are associated with a higher risk of RPL. The basic investigations of a couple affected by RPL should also include the evaluation of maternal anatomical anomalies (uterine leiomyoma, Müllerian anomalies and uterine synechiae) and maternal comorbidities (uterine anomalies, thrombophilic disorders, endocrinopathies, hormonal disorders, metabolic disorders and infections) [7], though not forgetting the evaluation of the male side, in particular the rate of sperm DNA fragmentation and, more in general, sperm quality [8]. Additionally, lifestyle factors, such as the alteration of BMI, smoking or excessive drinking, seem to play a key role in couples affected by RPL [6,9]. Nevertheless, in more than 50% of women, the cause of RPL remains undiagnosed, and the probability of a live birth in women with a history of RPL seems to range from 42 to 86% [10].

The aim of this review is to illustrate the possible pathological mechanisms involving HLA-G in RPL, preceded by an overview of the physiological context in which these mechanisms are involved. Therefore, in the first part of the review, the HLA-G structure, the placentation process and the HLA-G immunological role during pregnancy will be explained. In addition, the available evidence demonstrating the HLA-G molecule’s role in RPL will be analyzed.

## 2. HLA-G Structure and Receptors

The Major Histocompatibility Complex (MHC) was first discovered by Peter Gorer in 1936, and only thanks to his discovery, transplants have become possible [11]. MHC is a genomic region located on the short arm of chromosome 6 that includes more than 200 genes encoded for HLA molecule complexes. Two MHC classes are recognized: MHC class I and MHC class II. Both class molecules can present antigens: MHC class I to CD8+ T lymphocytes and MHC class II to CD4+ lymphocytes. The difference between the two classes is that MHC class I molecules can present intracellular peptides derived from cytosol or the nucleus, and MHC class II can present exogenous peptides. MHC class II molecules are encoded by three polymorphic genes (HLA-DR, HLA-DQ and HLA-DP) and are expressed by professional APCs (antigen-presenting cells) such as DCs (dendritic cells), macrophages and B cells [12]. The MHC class I includes classical class I genes (HLA-A, HLA-B and HLA-C) and non-classical genes (HLA-G, HLA-E and HLA-F) encoding for molecules with mainly immunomodulatory functions. HLA-G is considered a non-classical MHC molecule because it presents differences compared to classical MHC molecules. Classical MHC I molecules are expressed on all body cells’ surfaces, whereas in physiological conditions, HLA-G is expressed by a limited number of tissues, such as the thymus, cornea, nail matrix, stem cells and trophoblast. HLA-G exons encoding for peptide-binding domains show extremely low polymorphism when compared to the corresponding sequences in the HLA-A, HLA-B, and HLA-C genes, which are highly polymorphic. Nevertheless, the HLA-G gene has a distinctive 5′ upstream regulatory region (5′URR) and a high variability of 3′ untranslated regions (3′ UTR), that may affect surface HLA-G expressions [13]. With mRNA-alternative splicing, seven HLA-G protein isoforms can be generated, displaying a high diversity of molecular structures. HLA-G molecules are formed by a heavy chain containing alfa domains and a β2-microglobulin. HLA-G has seven introns and eight exons: exons 2, 3 and 4 encode for HLA-G alfa domains alfa 1, alfa 2 and alfa 3, exon 5 encodes for the transmembrane domain, and exons 7 and 8 are not translated. Gene encoding for the β2-microglobulin is located on chromosome 15. HLA-G has membrane-binding and soluble isoforms: HLA-G1, HLA-G2, HLA-G3 and HLA-G4 are membrane-binding proteins and HLA-G5, HLA-G6 and HLA-G7 are soluble proteins. HLA-G isoforms can form homodimers, increasing their affinity for the receptors (Figure 1). HLA-G receptors are CD8, LILRB1, LILRB2 and KIR2DL4 (CD158d). LILRB1 (Leukocyte Immunoglobulin-Like Receptor B1) is expressed by monocytes, B-cells, dendritic cells and natural killers. LILRB2 is expressed only by monocytes, and KIR2DL4 (killer cell immunoglobulin-like receptor 2DL4) is typically expressed by placental natural killers [14].

## 3. Placentation

Placentation is the process leading to placental development and represents the most challenging immunological step during pregnancy. At this time, fetal antigens and maternal immune system interact for the first time, and in physiological pregnancies, a delicate balance is reached between immunotolerance and the containment of trophoblastic invasion to the endometrial tissue. The human placenta differs from that of any other species and exhibits unique features. Both humans and rodents possess hemochorial placenta, which means that fetal trophoblasts are in direct contact with maternal blood. However, unlike mice that have a labyrinthine placenta, the human placenta is composed of chorionic villi submerged in maternal blood that fills the intervillous space [15]. Placentation starts on the eighth or ninth day after conception, when the blastocyst takes contact with the decidualized uterine mucosa. At this stage, two populations of cells can be identified: the mass of inner cells and the trophectoderm cells in the outer layer of the blastocyst. Placental tissue originates from trophoblast cells, which are derived from trophectoderm. Trophoblast cells include syncytiotrophoblast cells, an irreversibly differentiated tissue, and cytotrophoblast cells. Cytotrophoblast can still differentiate and give rise to villous cytotrophoblast cells, forming the multinucleated epithelium of the villi responsible for nutrient exchange and hormone production, and extravillous cytotrophoblast cells (EVTs). The latter are the cells responsible for placental anchoring to the decidua and for the vascular invasion process [16]. Through this process, the spiral arteries’ remodeling is achieved: the spiral arteries mutate into low-resistance and high-capacity vessels, reaching a 5–10-fold dilation [17]. This is an amazing phenomenon typical of mammals and in particular of humans. EVTs invade spiral arteries’ vascular lumen and replace the smooth muscle cells as far as the inner third of the myometrium, but blood flow reaches the intervillous space only at the end of the first trimester (Figure 2). Before this moment, in fact, the invasion of massive EVTs causes fetal cells to aggregate within the vasal lumen and arterial obstruction. Thus, in the first trimester of pregnancy, a transient condition of hypoxia is established, which is a function in correct fetal and placental development [18].

Placentation is a complex process in which foreign cells invade vascular structures of an organ, growing and proliferating without, in physiological conditions, either being rejected by the immune system or going beyond uterine limits. This consideration leads to compare placentation into a semiallograft, established and maintained naturally without the need for medical immunosuppression, therefore representing a fascinating immunological paradox unique in its kind.

## 4. HLA-G Role in Physiological Pregnancy

HLA-G is an important factor in the maternal acceptance of the fetus, playing a central role in the maintenance of an immunosuppressive state and in spiral artery remodeling [19,20]. Thus, HLA-G has been extensively studied in placentation disorders such as pre-eclampsia and fetal growth restriction [21].

The study of trophoblast organoids by Turco and coll. proved that EVTs are the trophoblast cells that most express HLA-G. EVTs are also the cells showing a greater adhesion in vitro: HLA-G+ cells can migrate out of the organoids, digest the matrigel to form tracks and eventually adhere to the surface of the culture [22]. In coherence with this finding, it is not surprising that HLA-G is a marker for gestational trophoblastic tumors due to its invasiveness [23].

As mentioned above, the generation of a transient hypoxic state is a normal phase of physiological placentation, and hypoxia has been shown to be a strong inducer of HLA-G expression in trophoblast [24] and in tumor cells [25]. However, Mori and coll. showed that in human trophoblast under prolonged hypoxic conditions, inhibitor factors targeting HLA-G 3′ UTR are upregulated and can repress HLA-G expression [26]. These apparently opposite results may suggest that hypoxia is actually useful to maintain a balance in trophoblast proliferation [27].

Immunomodulation is the result of cross talk between EVTs and maternal immune system cells, among which the most relevant are Natural Killer cells (NKs). NKs are cytotoxic innate lymphoid cells representing only 5–15% of peripheral blood lymphocytes and are so called for their spontaneous cytotoxicity, discovered for the first time in the context of cancer [28]. NKs represent 70–80% of the total placental leukocyte population at the beginning of pregnancy and decline in concentration during gestation. Compared to peripheral blood NKs, placental NKs (pNK) display higher expressions of inhibitory receptors, mainly KIR2DL4 and LILRB1, and show lower cytotoxic activity [29,30]. It has been proved that beginning from the menstrual cycle’s proliferative phase and throughout the pregnancy, NKs circulating in peripheral blood distribute concentrically around spiral arteries attracted by CXCL10 and CXCL11 endometrial expressions induced by progesterone and estrogen. [31,32,33]. NKs surround spiral arteries coming in close contact with EVTs. In this way, NKs and EVTs exchange signals with each other. pNKs can stimulate EVTs to invade and produce IL-8, which increases MMP-9 secretion and reduce EVTs’ apoptosis, but can also balance EVTs’ invasion by expression of TNF-α, TGF-β and IFN-γ, which inhibit the excessive proliferation of EVT [34]. On the other hand, HLA-G interacts with LILRB1, KIR2DL4 and CD8 on the NK’s surface, inhibiting cell proliferation, cytotoxicity and chemotaxis as well as inducing protective factors’ secretions, such as IL-10 [35]. Further, immunotolerance is also the result of HLA-G interaction with the other immune system cells located in the decidua, although in a lower number compared to NKs (e.g., macrophage, B cells and dendritic cells) [36].

IFN-gamma has been already described as a key player in implantation and placental development in mice [37]. It can be argued that IFN-gamma is usually considered a pro-inflammatory cytokine; however, due to the interaction with IL-10, progesterone and cAMP, it can promote an immune-tolerant profile that is fundamental for normal pregnancy.

In 2008, Blanco et al. [38] demonstrated that IFN-gamma enhances the expression of HLA-G, and this mechanism might be involved in the attempt of decidual stromal cells to block the cytotoxic attack of the decidual NK cells they have previously activated. Of note, decidual stromal cell apoptosis is described as a normal process in physiological pregnancy [39].

Furthermore, dysfunction of decidual NK can be detrimental for pregnancy, leading to recurrent pregnancy losses and placentation disorders [40]. In these specific pregnancy complications, previous studies showed altered cytokine profiles, in particular of VEGF-A, IFN-gamma and TNF-alfa [41]. However, the exact pathogenic mechanism has not yet been unraveled (see the paragraph “HLA-G in recurrent pregnancy loss”).

The relationship between EVTs and pNKs seems to go beyond the simple exchange of signals. The pNKs can acquire EVTs’ immunomodulatory properties through HLA-G trogocytosis. Trogocytosis is a phenomenon that allows membrane proteins to transfer between cells, enabling immune system cells to acquire proteins produced by other cells through endocytosis on their surface. HLA-G trogocytosis can give the receiving cell an immunosuppressive phenotype, and it has been proved that pNKs express HLA-G on their surface without possessing HLA-G mRNA [42,43,44].

Through its soluble isoform (sHLA-G), HLA-G can also perform systemic immunomodulatory effects. sHLA-G is produced in the very first stages of embryo development, and it can be detected in embryo culture medium, where its concentration seems to be predictive of successful implantation after IVF procedures [45]. Morandi and coll. proved that sHLA-G downregulates the expression of chemokine receptors in vitro on T cell populations isolated from peripheral blood, and that downregulation can be reverted by adding an antibody blocking sHLA-G receptor LILRB1 [46]. The main clinical studies on sHLA-G have been carried out in oncological diseases. In many tumors, increased sHLAG concentration correlates with higher metastasis rate and a worsening prognosis, and that confirms sHLAG’s potential immunosuppressive properties [47].

EVTs can affect the maternal immune system remotely not only by producing sHLAG, but also by shedding extracellular vesicles (EVs) in blood flow. EVs are phospholipid bilayer-enclosed vesicles that are quite heterogeneous in size, can be released by different cells, and can carry various types of antigens, cell surface receptors and ligands [48,49]. Kshirsagar and coll. demonstrated that exosomes isolated from first-trimester villous explant cultures and cytotrophoblast cells carry HLA-G molecules [50]. It has been demonstrated that EVs may have a role as modulators of embryo implantation [51] and affect the maternal immune system, contributing to the maintenance of an immunosuppressive phenotype [52]. Indeed, Hedlund and coll. showed that human placenta-derived EVs bearing MHC class I chain-related (MIC) proteins reduced cytotoxicity-binding NKG2D receptors on NK cells and cytotoxic T cells [53]. As for sHLA-G, HLA-G-EV’s role in tumor immune escape is well-described [54] (Figure 3).

## 5. HLA-G in Recurrent Pregnancy Loss

According to ESHRE guidelines, RPL is defined as the loss of two or more pregnancies [4]; its prevalence, probably underestimated, is 1–4% of all women who achieve pregnancy, and about 40–60% percent of RPL cases are unexplained [55]. HLA-G expression is thought to be involved in unexplained RPL pathogenesis. The great interest in HLA-G is due to its potential use as a clinical marker for adverse pregnancy outcomes, including miscarriage, RPL and recurrent implantation failure (RIF) [56]. The clinical interest focuses both on sHLA-G concentration determination and on HLA-G gene polymorphism research.

Regarding sHLA-G determination as a potential miscarriage marker, only a few studies have been conducted. In a longitudinal study, Alegre and coll. proved that sHLA-G increases in pregnant women. They compared the sHLA-G concentrations, measured using Enzyme-Linked Immunosorbent Assays (ELISA), of 45 healthy pregnant women with the sHLA-G concentrations of 14 healthy age-matched nonpregnant women. sHLA-G concentration was significantly higher in pregnant women in any of the three trimesters of pregnancy than in nonpregnant women. It is important to note that the authors found a wide intraindividual variation of sHLA-G concentrations during pregnancy. Among these 45 women, eight women included initially in the study had spontaneous miscarriages before 12 weeks of gestation. In the plasmatic sample obtained before spontaneous miscarriage, the authors found significantly lower plasmatic sHLA-G concentrations than the plasmatic sHLA-G obtained from pregnant women at the same gestational age who had a successful pregnancy afterward. They also found that in the women who had spontaneous miscarriages, the plasmatic sHLA-G concentration was lower than even that obtained from the control group of nonpregnant women [57]. Zidi and coll. dosed sHLA-G using ELISA in 55 nonpregnant women with a history of spontaneous miscarriage, 11 of whom had a history of RPL, and compared them with 56 healthy nonpregnant women and with 108 healthy women at different stages of uncomplicated pregnancy, though the trimester of pregnancy at sample collection is not indicated. They found no difference between the group of women with a history of spontaneous miscarriage and the healthy nonpregnant women group. Although without significance, the authors found lower sHLA-G levels in the RPL subgroup compared to the subgroup of women with only one miscarriage. As expected, they found lower levels of sHLA-G in the miscarriage group compared to pregnant women [58]. In this study, it is not indicated whether RPL patients underwent a diagnostic examination to exclude other RPL causes. In the HLA-G analysis of RPL patients, it should be verified if other RPL causes were excluded. This is particularly important for rheumatological diseases associated with RPL because HLA-G expression seems to be involved also in Systemic Lupus Erythematosus (SLE) and Antiphospholipid Syndrome (APS) pathogeneses. In fact, SLE patients seem to have significantly higher sHLA-G plasma levels if compared with healthy people [59]. Furthermore, De Carvalho and coll. showed that sHLA-G levels increased in APS patients and that, in these patients, heparin was a strong sHLA-G inducer. They also observed that patients under heparin treatment who had obstetric events had increased levels of sHLA-G compared to patients who had vascular events [60]. Therefore, to not incur possible bias, known causes of RPL should be excluded a priori. Moreover, pregnant patients should be always grouped and compared by carrying out gestational age-matching because sHLA-G values fluctuate during trimesters of pregnancy, decreasing from the second trimester onwards [61]. Krop and coll. have analyzed sHLA-G in different trimesters of pregnancy. They compared the sHLA-G blood concentrations of 22 pregnant women in their first trimester with RPL history and a normal pregnancy outcome with the sHLA-G blood concentrations of nine women with RPL with first-trimester pregnancy loss, and they found no significant difference. However, they found that in women with a RPL history and normal pregnancy outcome, sHLA-G blood concentrations were not significantly increased during the 3rd trimester compared to nonpregnant control patients, whereas sHLA-G blood concentrations in 3rd-trimester control pregnancies without a history of RPL increased compared to nonpregnant control patients [62]. Different results have been obtained by Maddaru and coll., who recruited 135 pregnant women with a history of RPL and 135 age-matched healthy, parous pregnant women. They excluded women with a history of only one spontaneous miscarriage, induced abortions and miscarriages with known reasons. They analyzed with ELISA soluble HLA-G5 isoform serum concentrations and found them higher in control patients when compared to RPL patients; this difference was also found when they divided the patients according to the trimester and compared first-trimester RPL patients to first-trimester control patients and second-trimester RPL patient to second-trimester control patients [63]. HLA-G protein expression was also studied on placental tissue. Craenmehr and coll. evaluated both HLA-G protein and HLA-G mRNA levels in at-term placentas of RPL patients and healthy control patients. They found no differences between HLA-G trophoblast tissue expression between the groups. However, they found unexpectedly higher HLA-G expression on the decidual tissue of RPL patients compared to control patients [64]. Regarding the expression of HLA-G at the trophoblast level, because these are two groups of patients with normal full-term pregnancies, it may be normal to expect regular trophoblastic development and therefore an equal expression of HLA-G. It is technically difficult investigating the changes in each RPL patient that allowed a successful pregnancy. The patient may have been under medical treatment during pregnancy, not performed in the previous pregnancies, and, as we will explain next, some medications may induce HLA-G expression. It should be also considered that HLA-G upregulation could be seen not as the consequence of RPL history but as the source of a successful obstetrical outcome after RPLs.

Over the years, researchers also really focused on the evaluation of possible associations between HLA-G polymorphisms and unexplained RPL, especially when studying 3′ untranslated regions (3′ UTRs) and the 5′ upstream regulatory region (5′ URR) [65]. In particular, 3′ untranslated regions (3′ UTRs) are mRNA regions that are not translated in proteins, but that seem to have a regulatory role controlling mRNA stability, mRNA translation, mRNA localization and spatial organization of protein production. In fact, microRNAs (miRNAs), which are short, non-coding, single-stranded RNA molecules, can target 3′ UTRs modulating gene expression through mRNA degradation and translation blockage [66]. The 3′ UTR is a highly polymorphic region considered to be located mostly in HLA-G exon 8. The most studied 3′ UTR polymorphism consisting of a 14-nucleotide deletion, also known as the 14-bp indel (insertion/deletion) polymorphism, is characterized by the removal of a 14-nucleotide segment [67]. This polymorphism has been associated with lower membrane-bound HLA-G and sHLAG production [68]. Regarding the possible association between 14-bp indel polymorphism and risk of RPL, in a recent meta-analysis including ten studies, the prevalence of HLA-G 14 bp polymorphism in homozygosis (insertion/insertion) and heterozygosis (insertion/deletion) was investigated in 1091 women with RPL and 808 women without RPL. Women with women with RPL had a higher prevalence of the HLA-G 14 bp insertion/insertion genotype, whereas HLA-G ins/del 14 bp genotype prevalence was not statistically significantly different between the two groups [69]. In contrast, Kalotra and coll. did not find significant differences in the allele frequencies of the HLA-G 14 bp insertion/deletion polymorphism between women with RPL and women without RPL but proved that sHLA-G concentration was significantly lower in women with RPL and with the 14 bp heterozygous genotype compared to the control group with the same genotype [70].

Other polymorphisms that can be found in HLA-G 3′ UTR are single nucleotide polymorphisms (SNPs) that may affect miRNA binding sites, increasing the affinity of specific miRNAs blocking HLA-G mRNA and decreasing HLA-G expression. miRNA’s role in controlling HLA-G expression has been studied by Zhu and coll. on JEG-3 choriocarcinoma cells, a widely used tool employed as a model for placental trophoblasts [71]. They proved that miRNAs affect HLA-G expression, which determines lower HLA-G mRNA and protein levels on JEG-3 cells transfected with hsa-miR-152. They also demonstrated that hsa-miR-152 interacts with the 3′-UTR of the HLA-G mRNA [72]. It was hypothesized that HLA-G 3′ UTR SNPs can be associated with increased risk of RPL. Michita and coll. found associations between 3′UTR SNP (+3010CC, +3142GG, +3187AG) and RPL, and they did not observe a significantly higher prevalence of 14 bp ins/del polymorphism in the RPL group but saw a tendency towards a higher proportion of homozygosis (insertion/insertion) within this group [73]. In addition, Bai and coll. found no significant difference in the frequency of the 14 bp ins/del polymorphism between RPL patients and the control group, but they did observe that the +3010CC genotype could be a risk factor for RPL, whereas the +3187GG genotype might be a protective factor [74].

In the HLA-G gene as well, the promoter region may show variability, that can influence HLA-G levels by modifying the binding affinity for transcription factors. HLA class I genes present two main regulatory regions: Enhancer A (EnhA), which interacts with the NF-κB family, and the interferon-stimulated response element (ISRE). As mentioned at the beginning, the HLA-G 5′ upstream regulatory region (5′URR) has some distinctive features: HLA-G EnhA has low affinity for NF-κB, and HLA-G ISRE does not mediate IFN-γ-induced transactivation [75]. HLA-G 5′URR polymorphism may have an impact on HLA-G gene expression, modulation, and on sHLA-G blood levels [76], and therefore, it can be associated with adverse pregnancy outcomes. Agrawal and coll. screened 283 RPL patients finding 35.34% (n = 100) with unexplained RPL. Among these patients, -1179G > A, -725C > G/T and -486A > C SNP carriers showed an increased risk of RPL. Regarding HLA-G levels, they measured HLA-G mRNA but not HLA-G proteins, finding significantly decreased HLA-G mRNA levels in -1179G > A and -725C > G/T SNP carriers. Interestingly, Agrawal’s research group also studied paternal HLA-G 5′URR polymorphism and parental polymorphism associations, finding that an increased 3.5-fold RPL risk was associated with a parental -1179G > A SNP combination of AA x AA and that a 4.3-fold RPL risk was related to a parental -725C > G/T SNP combination GG x GG. As expected, couples in which both partners were homozygous for the mutant allele showed a higher RPL risk [77]. Tang et al. demonstrated that besides the promoter and the 3′UTR polymorphisms, improper DNA methylation of HLA-G genes may play a role in dysregulation during pregnancy, especially in pre-eclamptic women [78]. Currently known HLA-G mechanisms in physiological pregnancy and in recurrent pregnancy loss are summarized in Table 1.

The interest in HLA-G expression is also largely due to its possible therapeutic implications. Progesterone is well known for its immunosuppressive effects in pregnancy, and several studies demonstrate that progesterone can induce HLA-G expression. Progesterone’s effect was observed both in the stem cells of different tissues [79] and in JEG-3 choriocarcinoma cells and cytotrophoblasts. JEG-3 choriocarcinoma cell and human cytotrophoblast cell cultures incubated with progesterone show higher HLA-G mRNA and HLA-G protein expression. Adding mifepristone, a progesterone antagonist, to the cultures decreased HLA-G mRNA and HLA-G protein levels, confirming that progesterone’s effect resulted from a direct receptor binding [80]. Akhter and coll. compared trophoblast cell cultures isolated from the placental tissue of women with RPL with trophoblast cell cultures isolated from placental tissue undergoing induced first-trimester termination of normal pregnancies. They observed increased HLA-G mRNA levels in cultured RPL trophoblast cells incubated with glucocorticoids (dexamethasone and hydrocortisone) [81]. Another interesting molecule that can influence HLA-G expression is Preimplantation Factor (PIF), a small peptide secreted by vital embryos with pleiotropic effects promoting pregnancy. In JEG-3 choriocarcinoma cells, PIF can enhance both intracellular and surface HLA-G expression [82]. No study has yet been conducted on the effect of heparin, widely prescribed in RPL patients during pregnancy [83], on the trophoblast HLA-G expression.

As we mentioned above, HLA-G may be also carried on extracellular vesicles (EVs). The role of EVs was largely studied in relation to the pathogeneses of pregnancy disorders, especially in pre-eclampsia and fetal growth restriction [84,85]. In particular, pre-eclamptic women showed an aberrant expression of HLA-DR on syncytiotrophoblast-derived extracellular vesicles [86,87]. Concerning the role of EVs in the pathogenesis of RPL, few studies have been conducted. In an abortion-prone murine model, the secretion of EVs was inhibited, and the intravenous transfer of exogenous EVs carrying HLA-E gave a protective effect on pregnancy outcomes; therefore, it promoted secretion of IFN-γ and VEGFα with NK cells [88]. HLA-G and HLA-E both bind the NKG2D receptor [89]; this evidence may suggest that an altered secretion of EVs carrying HLA-G can be involved in recurrent pregnancy loss’s pathogenesis. In a recent prospective cohort study, pregnant RPL patients were recruited, nine of whom had pregnancy loss, whereas twenty-five had a live birth. EVs circulating in peripheral blood were analyzed for CD4, CD9, CD45, HLA DRP/DQ/DR, HLA G, FasL, TRAIL and Hsp70 expression. CD9 was the only EV marker significantly increased in the live birth group. HLA-G was also higher in the live birth group compared with pregnancy loss group, but the difference was not statistically significant [90]. In a placebo-controlled trial, Jørgensen and coll. compared HLA-G EVs expression in the peripheral blood of two groups of pregnant women with RPL. Of these patients, 19 received intravenous immunoglobulin (IVIG) treatment, and 20 patients received a placebo. In the IVIG group, HLA-G EVs expression was significantly higher compared with HLA-G EVs expression of the placebo group [91]. The current evidence about the role of EVs carrying HLA-G in physiological pregnancy and in recurrent pregnancy loss is limited; the main findings are reported in Table 2.

## 6. Conclusions

The physiological role of HLA-G in implantation and placentation is still not completely understood, and over the years, new mechanisms and other immunomodulatory molecules, such as HLA-E and HLA-F, seem to play important roles as well [92].

Results regarding the possible use of sHLA-G plasma concentration as a risk factor for RPL in nonpregnant women are controversial because in nonpregnant women, sHLAG values are normally much lower due to the absence of trophoblastic cells.

s-HLAG plasma concentration during pregnancy seems to be a more promising tool as a miscarriage marker in women with RPL. Unfortunately, sHLA-G values vary widely between individuals and also have great variability during different trimesters of pregnancy. Moreover, sHLA-G concentrations may also be affected by other comorbidities, such as obesity [93].

Specific sHLA-G isoforms are not detected in all studies. Because HLA-G has large structural variety, it may be relevant to know if, in RPL patients without decreased total sHLA-G levels as compared to control patients, there are decreased levels of specific HLA-G isoforms. Finally, in several studies, the presence of HLA-G was assessed using only mRNA but not protein levels. That leads to obtaining only partial information about protein expression, considering the important post-transcriptional regulation at mRNA level in HLA-G expression and in HLA-G isoform production [94].

The studies of HLA-G maternal polymorphisms and of RPL risk have controversial results. The HLA-G male genotype is often poorly considered in studies, but it is important to consider that EVT’s protein expression pattern originated not only from the mother’s genetic heritage but also from the father’s; therefore, in these studies, the paternal genotype should always be considered. In addition, the analysis of parental genotypes should be correlated with trophoblast genotypes, trophoblast HLA-G expression levels and obstetric outcomes. More studies are required to understand if increasing HLA-G expression in unexplained RPL couples may be a potential therapeutic target to improve obstetrical outcomes.

## Figures and Tables

**Figure 1 ijms-24-02557-f001:**
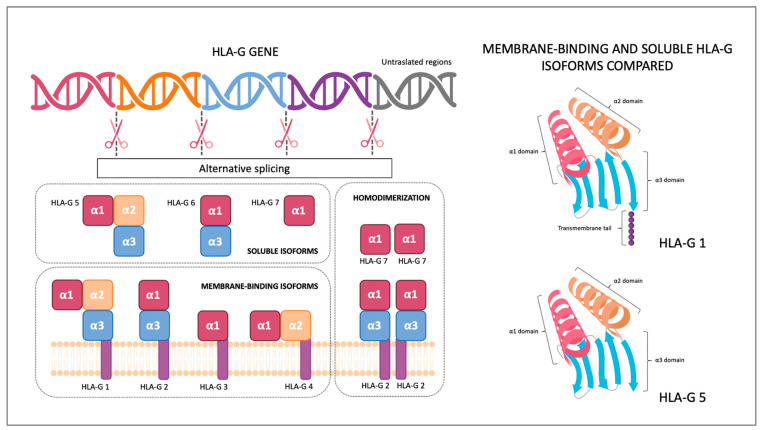
Human Leukocyte Antigen (HLA)-G isoforms. With mRNA-alternative splicing, seven HLA-G protein isoforms can be generated: HLA-G membrane-binding isoforms (HLA-G1, HLA-G2, HLA-G3 and HLA-G4) and HLA-G soluble isoforms (HLA-G5, HLA-G6 and HLA-G7). Both HLA-G membrane-binding isoforms and HLA-G soluble isoforms can form homodimers. As shown in the right side of the image, HLA-G 1 is characterized by a six amino acid-long cytoplasmic tail that anchors it to the cell membrane.

**Figure 2 ijms-24-02557-f002:**
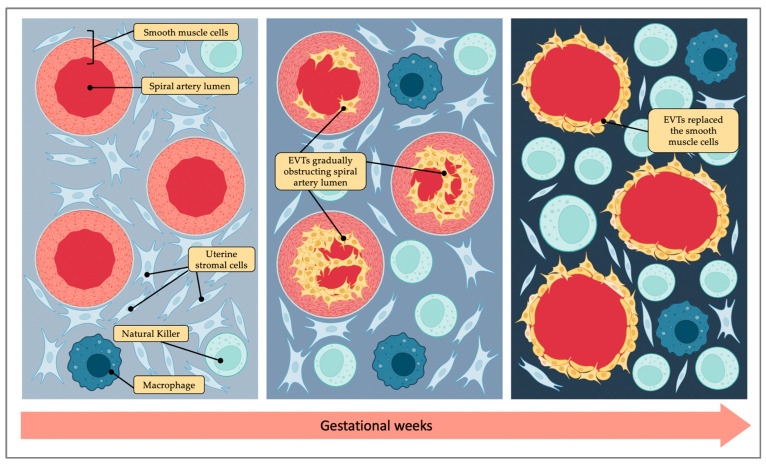
Spiral artery remodeling. Extravillous cytotrophoblast cells (EVTs) invade spiral arteries’ vascular lumen and replace the smooth muscle cells. In the first stage of this process, EVTs aggregate within the vasal lumen, causing arterial obstruction. Then, at the end of the first trimester, blood flow reaches the intervillous space, and spiral arteries mutate into low-resistance and high-capacity vessels, reaching a 5–10-fold dilation. At the same time, as gestational weeks go by, decidual is enriched with natural killers and macrophages.

**Figure 3 ijms-24-02557-f003:**
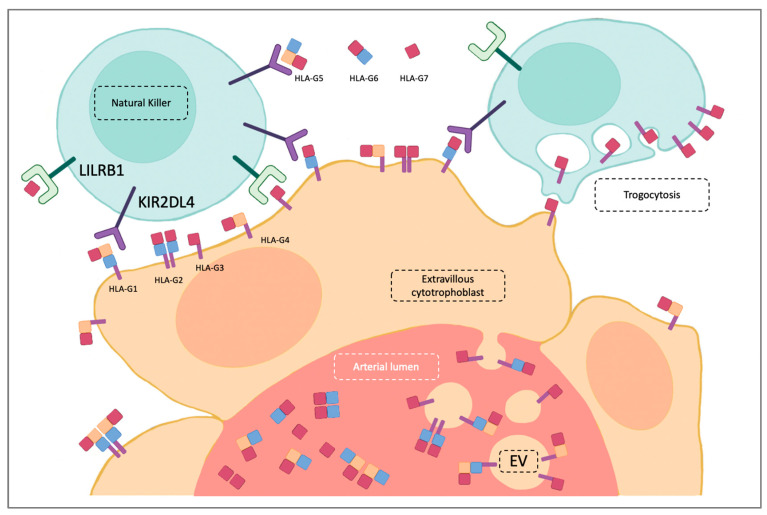
Extravillous cytotrophoblast cells (EVT) and natural killer (NK) interactions. HLA-G isoforms expressed on extravillous cytotrophoblast interact with LILRB1 and KIR2DL4 on NK surface. In addition, by trogocytosis, placental NKs (pNKs) can acquire HLA-G membrane-binding isoforms on their surface. Soluble HLA-G isoforms can be spread in circulation and in the extracellular matrix. In addition, membrane-binding HLA-G isoforms can be spread when carried on extracellular vesicles (EV).

**Table 1 ijms-24-02557-t001:** Human Leukocyte Antigen-G (HLA-G)’s roles and mechanisms in physiological pregnancy and in recurrent pregnancy loss.

Role of HLA-G in Physiological Pregnancy	Role of HLA-G in Recurrent Pregnancy Loss
HLA-G inhibits uterine natural killers’ cytotoxicity HLA-G inhibits CD8′s cytotoxicity HLA-G inhibits chemotaxis HLA-G induces natural killers’ secretion of IL-10 IFN-gamma enhance HLA-G expression, blocking the cytotoxic attack of decidual NK cells Through trogocytosis, natural killers can express HLA-G, acquiring immunosuppressive phenotypes	Lower plasmatic sHLA-G concentration Lower HLA-G membrane binding expression on trophoblast 3′ UTR polymorphisms (14-nucleotide deletion) may have an impact on HLA-G expression Promoter-region polymorphisms may have an impact on HLA-G expression DNA methylation of HLA-G gene may play a role in dysregulation during pregnancy Diminished expression of KIR2DL4 on placental natural killers [29]

**Table 2 ijms-24-02557-t002:** Extracellular vesicles carrying HLA-G roles in physiological pregnancy and in recurrent pregnancy loss.

Physiological Pregnancy	Recurrent Pregnancy Loss
Extracellular vesicles carrying HLA-G may bind NKG2D receptor on uterine natural killers, promoting IFN-gamma e VEGF-α secretion Extracellular vesicles carrying HLA-G interact with peripheral blood cells and reducing cytotoxicity	Altered secretion of extracellular vesicles Altered HLA-G expression on extracellular vesicles’ surfaces

## Data Availability

Not applicable.

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
