# Peer review of "HLA-G and Recurrent Pregnancy Loss"

_ijms, 2023, doi:10.3390/ijms24032557_

Round 1

Reviewer 1 Report

Title: HLA-G and recurrent pregnancy loss

Type: review

Authors: Greta Barbaro , Annalisa Inversetti , Martina Cristodoro , Carlo Ticconi , Giovanni Scambia , Nicoletta Di Simone *

Comments:

The review is well written and relatively well organized which fundamentally describes the role of HLA-G in the modulation of immune response during pregnancy and possible pathological mechanisms and clinical aspects involving HLA-G in the disruption of the immunologic balance during pregnancy thus increasing the risk of recurrent pregnancy loss. The review manuscript fits adequately with the purpose of IJMS journal as it describes in detail the HLA-G genes structure and activity and particularly their function during the gestation period. In spite several reviews have already published in the field, in my opinion, the present manuscript could improve our knowledge and it can be accepted for publication with minor corrections. The work is interesting and comprehensively describes the state of the art of HLA-G genes activities during a physiological pregnancy and possible dysregulations which have been related to recurrent spontaneous abortion. Figures are well made and appropriate and in my opinion do not require revision. I have some specific comments with some concerns and minor suggestions to further improve the manuscript.

a) My major observation is that given recurrent pregnancy loss is one of the main topics of the review manuscript, I suggest including a brief introductive section describing the general aspects of this disease, such as etiology, risk factors and epidemiology.

Please see below several minor observations which I believe would improve the quality of the manucript

a) In figure 1 caption, in order to improve the reading, I suggest including the complete gene names. For instance, “Human Leukocyte Antigen G (HLA-G)”. Please also check all figures’ captions

b) In line 131 I suggest replacing normal with physiological

c) The expression of MHC class-I have been reported to be dysregulated in cancer (PMID: 34768895). For completeness, this information should be included 

d) There is no mention in the text of various detailed reviews on the role of HLA-g during pregnancy which have already been published, some of those even quite recently. I suggest including these works PMID: 36532068, PMID: 33193435, PMID: 33745758, PMID: 35571013.

e) In the sentence in line 164, a reference should be included. For instance PMID: 31681264

f) In lines 267-269 references should be included as a support of the statement. May I suggest PMID: 33859649

g) When human miRNAs are mentioned, they should be named “hsa-miR”, for instance hsa-miR-152 (line 296).

h) Besides polymorphisms at the promoter and 3’UTr region, improper DNA methylation of HLA-G genes have bene reported to play a role in their dysregulation during pregnancy (preeclampsia) (PMID: 26116450. Authors should mention this point as a connection with abortive events cannot be excluded 

These are several minor typo errors I have found. 

a) In line 152, 156. 171 etc.. multiple citations should be merged

b) Line 273 better “which are..”

c) Please do not start a sentence with a number (line 274)

d)

Reviewer 2 Report

Manuscript ID: ijms-2164170

Title: HLA-G and recurrent pregnancy loss

Authors: Greta Barbaro , Annalisa Inversetti , Martina Cristodoro , Carlo Ticconi , Giovanni Scambia , Nicoletta Di Simone

This is a nicely written review describing alterations in the HLA-G pathway and potential links to disrupting the immunologic balance and increasing the risk of recurrent pregnancy loss in women. Overall, the review is interesting, however, several articles related to the structure of the figures and illustrations presented in this manuscript, as well as other edits and missing critical information would need to be addressed.

 Major concerns:

-       Figure 1: It would be most informative and beneficial should the authors include an illustration of the protein structure of the HLA-G and its soluble isoforms (sHLA-G) for a comparison. Current figure 1 is the least informative in its current format.

-       Current Figure 2 is the least informative and would need to be properly labelled and/or replaced. It’s unclear what cell types are being demonstrated in current Figure 2. Please adequately label all cell types presented in this figure, otherwise please consider replacing the current figure 2 with clearly labelled and readable infographics of the process of uterine spiral artery remodelling. Currently, the process of EVT invasion of the spiral arteries is very ill-demonstrated in the present figure 2.

-       Please provide tables summarizing the reported roles and suggested mechanisms of HLA-G in normal pregnancy and in RPL.   

-       Please further elaborate on the role of extracellular vesicles in the humoral and cellular immunodynamics of HLA-G in normal pregnancy and in RPL. Current discussion notes and arguments on extracellular vesicles and HLA-G in lines 342-354 need to be supported with further evidence. A table summarizing these findings is desirable.  

-       There is no mention of the role of interferon-gamma in the regulation and the immune cellular dynamics of the HLA-G in normal pregnancy and in RPL. Please consider adding a paragraph sufficiently addressing this critical concept of the regulation of the protein expression and cellular interactions of the HAL-G in normal human pregnancy and in RPL.   

Round 2

Reviewer 2 Report

Thank you for your revisions.